

# Is there genetic variation in mycorrhization of *Medicago truncatula*?

Dorothée Dreher[*], Heena Yadav[*], Sindy Zander and Bettina Hause

Department of Cell and Metabolic Biology, Leibniz Institute of Plant Biochemistry, Halle, Germany
[*] These authors contributed equally to this work.

## ABSTRACT

Differences in the plant's response among ecotypes or accessions are often used to identify molecular markers for the respective process. In order to analyze genetic diversity of *Medicago truncatula* in respect to interaction with the arbuscular mycorrhizal (AM) fungus *Rhizophagus irregularis*, mycorrhizal colonization was evaluated in 32 lines of the nested core collection representing the genetic diversity of the SARDI collection. All studied lines and the reference line Jemalong A17 were inoculated with *R. irregularis* and the mycorrhization rate was determined at three time points after inoculation. There were, however, no reliable and consistent differences in mycorrhization rates among all lines. To circumvent possible overlay of potential differences by use of the highly effective inoculum, native sandy soil was used in an independent experiment. Here, significant differences in mycorrhization rates among few of the lines were detectable, but the overall high variability in the mycorrhization rate hindered clear conclusions. To narrow down the number of lines to be tested in more detail, root system architecture (RSA) of *in vitro*-grown seedlings of all lines under two different phosphate (Pi) supply condition was determined in terms of primary root length and number of lateral roots. Under high Pi supply (100 μM), only minor differences were observed, whereas in response to Pi-limitation (3 μM) several lines exhibited a drastically changed number of lateral roots. Five lines showing the highest alterations or deviations in RSA were selected and inoculated with *R. irregularis* using two different Pi-fertilization regimes with either 13 mM or 3 mM Pi. Mycorrhization rate of these lines was checked in detail by molecular markers, such as transcript levels of *RiTubulin* and *MtPT4*. Under high phosphate supply, the ecotypes L000368 and L000555 exhibited slightly increased fungal colonization and more functional arbuscules, respectively. To address the question, whether capability for mycorrhizal colonization might be correlated to general invasion by microorganisms, selected lines were checked for infection by the root rot causing pathogen, *Aphanoymces euteiches*. The mycorrhizal colonization phenotype, however, did not correlate with the resistance phenotype upon infection with two strains of *A. euteiches* as L000368 showed partial resistance and L000555 exhibited high susceptibility as determined by quantification of *A. euteiches* rRNA within infected roots. Although there is genetic diversity in respect to pathogen infection, genetic diversity in mycorrhizal colonization of *M. truncatula* is rather low and it will be rather difficult to use it as a trait to access genetic markers.

Corresponding author
Bettina Hause, bhause@ipb-halle.de,
Bettina.Hause@ipb-halle.de

## INTRODUCTION

The arbuscular mycorrhiza (AM) represents a unique interaction between two eukaryotes, an obligate biotrophic fungus and its host plant, leading to an improved fitness of both interacting partners (*Bonfante & Genre, 2008*). AM fungi play an enormous role in terrestrial ecosystems due to their ubiquitous occurrence and their widespread interaction with plants. AM fungi, e.g., the species *Rhizophagus irregularis* (DAOM197198), belong to the kingdom of fungi and are classified to the subphylum *Glomeromycotina* within the phylum Mucoromycota (*Spatafora et al., 2016*).

The association between plant roots and AM fungi has proven to be an evolutionary successful strategy, since more than 80% of all terrestrial plant species live in symbiosis with AM fungi (*Schüssler, Schwarzott & Walker, 2001*). The host plant supplies the fungus with photoassimilates, which are metabolized to glycogen or lipids as energy storage (*Bago, Pfeffer & Shachar-Hill, 2000*). In turn, the AM fungus assists its host plant in acquisition of mineral nutrients and water (*Govindarajulu et al., 2005*; *Parniske, 2008*). Phosphate is one of the limiting nutrients for plant growth owing to its inaccessible form which has poor solubility and very slow diffusion. Roots affect the Pi concentration of the soil solution by active phosphate uptake, hence creating a Pi depletion zone around the root (*Hinsinger et al., 2005*). The extraradical fungal mycelium widely expands the Pi-depletion zone and due to the minor hyphal diameter, smaller soil pores can be exploited by a larger absorbing surface. Hence the symbiosis with AM fungi displays a powerful mechanism for plants to increase Pi availability.

The family of Fabaceae represents the third largest family of higher plants including more than 20,000 species and 700 genera (*Doyle & Luckow, 2003*). Due to their suitability for plant genomics, two species of this family, *Medicago truncatula* (*Rose, 2008*) and *Lotus japonicus* (*Handberg & Stougaard, 1992*) have been established as model plants mainly in order to get insights into agronomical important legume-microbe interactions. Key attributes of *M. truncatula* include its small, diploid genome consisting of two-times eight chromosomes with about 500 Mbp, its self-fertile nature and its rapid generation time. The genome of *M. truncatula* was sequenced capturing 94% of all *M. truncatula* genes (*Tang et al., 2014*; *Young et al., 2011*). Moreover, numerous ecotypes of *M. truncatula* have been collected throughout the Mediterranean Basin, and the considerable phenotypic variation for features such as growth habit, flowering time, trichome formation, and disease resistance represents an important resource to examine the genetic basis of legume functions (*Bonhomme et al., 2014*; *Kang et al., 2015*; *Stanton-Geddes et al., 2013*).

Previously, most studies in respect to interaction of *M. truncatula* with AM fungi have focused on single reference lines or a limited number of populations. The *M. truncatula* ecotype collection used in the present work is based on a survey about the genetic diversity in a collection of 346 inbred lines spanning the bulk of diversity that has been collected throughout the species range to date (*Ellwood et al., 2006*). Thirteen microsatellite markers were used to assign genetic relationships and select a nested core collection of 32 inbred lines, representing the genetic diversity of the complete collection (*Ronfort et al., 2006*). This collection has already been used successfully to identify key regulatory genes,

which are involved in the resistance of *M. truncatula* to the root pathogen *Aphanomyces euteiches* (*Badis et al., 2015*; *Bonhomme et al., 2014*). *A. euteiches* belongs to the kingdom of Chromalveolata and the class of Oomycota and causes the root rot disease in legumes. Primary disease symptoms in infected roots are water-soaked, softened brown lesions followed by significant reductions of root mass. Secondary symptoms like chlorosis, necrosis and wilting of the foliage might follow (*Hughes, Teresa & Grau, 2014*).

Besides the search for resistant ecotypes followed by proper breeding strategies, an alternative to reduce damage caused by soil-born plant pathogens could be the application of AM fungi as they have been shown to induce resistance towards root pathogens, as e.g., *A. euteiches* (*Azcón-Aguilar & Barea, 1996*; *Hilou et al., 2014*; *Whipps, 2004*). Additionally, the interaction of plants with AM fungi results in further beneficial effects, such as increased uptake of nutrients like phosphate, nitrogen (ammonium and nitrate), zinc, copper and potassium (*Cavagnaro, 2008*), and can lead to an increase in plant growth rate and total plant biomass (*Harrison, 1999*; *Parniske, 2008*). Moreover, AM can improve the tolerance of the plant to certain abiotic stresses, including drought, salt, and heavy metals (*Kamel et al., 2017*). Therefore, identification of traits causing a well-established mycorrhiza might help to improve the overall plant fitness.

The aim of this work was, therefore, to evaluate the SARDI core collection of *M. truncatula* ecotypes, collected in different parts of the world (*Ronfort et al., 2006*), regarding their symbiotic interaction with *R. irregularis*. The colonization rate of all ecotypes was evaluated, either using highly active inoculum or native sandy soil. To narrow down the number of ecotypes in order to have a closer look to slight differences in their mycorrhization, ecotypes were selected according to differences in the root system architecture (RSA) of seedlings. RSA describes the spatial arrangement of roots in the soil or growth media, is often quantified in terms of length of the primary root and number of lateral roots (*Chevalier et al., 2003*) and is known to be influenced by the availability and distribution of nutrients in the soil (*Thaler & Pagès, 1998*). On the one hand, RSA is altered by phosphate, which can affect the primary root length and number of lateral roots (*Gruber et al., 2013*; *Kellermeier et al., 2014*), on the other hand, colonization of roots by AM fungi is dependent on RSA, but also influences RSA (*Gutjahr & Paszkowski, 2013*; *Hodge et al., 2009*). Moreover, the relation between AM and RSA became obvious by the fact that plant phosphate transporter genes expressed specifically in arbuscule-containing cells play also a role in regulating relevant developmental programs, like root branching (*Volpe et al., 2016*). Using two regimes of Pi fertilization, the lines most deviating in RSA were selected and again analyzed in more detail for the interaction with *R. irregularis*, but also regarding their susceptibility towards the root pathogen *A. euteiches*.

# MATERIAL & METHODS

## Plant material and growth conditions

Plant materials used in this study included the set of nested core collection of *Medicago truncatula* (L.) Gaertn. consisting of 32 lines (*Ronfort et al., 2006*) and var. Jemalong A17 (Table S1). Seeds were treated with concentrated sulphuric acid for 5 min followed by

intensive washing with distilled water. Seeds were subsequently placed on sterile glass petri dishes with filter paper and stored for three days at 4 °C in the dark. Germination was performed in the dark at RT for one day and in light at RT for another day. Seedlings were transferred into pots (one seedling per pot with a diameter of 13 cm) filled with 600 ml expanded clay of 2–5 mm particle size (Original Lamstedt Ton; Fibo ExClay, Lamsted, Germany). For inoculation with *Rhizophagus irregularis*, expanded clay was mixed with inoculum (see below), for infection with *Aphanomyces euteiches* pure expanded clay was used. Growth of plants in soil containing lower spore density and higher AM fungi diversity was performed in native soil from Großbbeeren (Germany). This sandy soil contained 1.3% organic matter, 4% of clay in the soil dry weight, and 15 mg extractable P/g soil and had a pH of 7.6.

All lines were grown in a phytochamber with a 16 h/8 h cycle (22 °C/18 °C) at 220 µmol photons m$^{-2}$ s$^{-1}$. Plants were watered with deionized water three times per week and fertilized weekly with 10 ml 10× Long Ashton solution (*Hewitt, 1966*) containing either 20% (corresponding to 3 mM) or 100% (corresponding to 13 mM) phosphate.

## Inoculation with *Rhizophagus irregularis*, plant harvest, and determination of mycorrhization rate

*R. irregularis* (Schenk and Smith, isolate 49; *Maier et al., 1995*) was enriched in propagules by co-cultivation with leek (*Allium porrum*, cv. Elefant) in expanded clay as described previously (*Schaarschmidt et al., 2007*). Leek inoculum containing *R. irregularis* hyphae and spores was carefully mixed with clean expanded clay in a ratio of 2:8 (v/v) and used as highly active inoculum.

Plants of all lines inoculated with *R. irregularis* were harvested at 21, 35 and 50 days after inoculation. Plants grown in sandy soil were harvested at 35 days. Plants were quickly removed from the pot, roots carefully separated from the substrate, washed with distilled water and dried with a paper towel. For determination of mycorrhization rate using the Gridline intersection method (*Giovannetti & Mosse, 1980*), the whole root was used for subsequent staining. Selected lines were analyzed in independent experiments using the calculation method according to *Trouvelot, Kough & Gianinazzi-Pearson (1986)*. Here, an approximately 2 cm wide, centrally located section of every root system was used for staining and the remaining root material was snap frozen in liquid nitrogen and stored at −80 °C until isolation of RNA. Staining of all mycorrhizal samples was done with 5% (v/v) ink (Shaeffer Skrip jet black, Sheaffer Manufacturing, Madison, WI, USA) in 2% acetic acid as described before (*Vierheilig et al., 1998*). Fungal structures were assessed using a stereomicroscope.

## Cultivation of plants for determination of root system architecture (RSA)

Seeds treated as described above were germinated on plates containing 0.7% plant agar (Duchefa Biochemie, Haarlem, North Holland, The Netherlands) at 12 °C in the dark for two days. In each case, seven seedlings were then transferred to plates containing Modified Strullu Romand Medium (*Declerck, Strullu & Fortin, 2005*) solidified with 0.4% (w/v) phytagel (Sigma-Aldrich, Munich, Germany) and either supplied with 100 µM phosphate

(high Pi) or 3 μM phosphate (low Pi). Plates were incubated vertically at 17−20 °C for seven days with a day/night cycle of 16 h/8 h. After taking photographs of all plates, length of primary roots and number of lateral roots were determined using the software 'Smart Root' (*Lobet, Pagès & Draye, 2011*). For each line and treatment at least three plates with seven seedlings each were evaluated.

## Cultivation of *A. euteiches* and infection of plants

The cultivation and production of zoospores of *A. euteiches* (Drechs.) strain AERB84 (kindly provided by Dr. Anne Moussart, INRA France) and strain GB I1 (kindly provided by Dr. Phillip Franken, IGZ Germany) were performed as described before (*Hilou et al., 2014*). Both strains were cultivated on CMA-HST plates consisting of 17 g l$^{-1}$ corn meal agar, 4 g l$^{-1}$ yeast extract, 0.8 mg l$^{-1}$ β-sitosterol, and 100 mg l$^{-1}$ α-tocopherol acetate in 50 mM phosphate buffer (pH 6.8–7.0) in the dark at RT for 4–5 days. When the entire surface of the plate was occupied by the mycelium, 1 cm$^2$ pieces containing young hyphae were cut and transferred into new Petri dishes. After addition of a solution consisting of yeast extract tryptone (3.5 g l$^{-1}$) and autoclaved lake water in a ratio of 3:1 (v/v), plates were incubated at RT in the dark for 2 days. The newly formed mycelium was rinsed three times with autoclaved tap water for 45 min followed by overnight incubation to induce zoospores. Seedlings of *M. truncatula* cultivated on vertical plates as described above were infected by adding of 200,000 motile zoospores to each root. Three weeks after infection, phenotypic alterations were evaluated. *M. truncatula* plants grown in expanded clay for two weeks were infected at the stem base by the addition of about 1,000,000 motile zoospores of *A. euteiches* per plant. Control plants were mock-inoculated with an equal volume of autoclaved water. One day before infection, all pots were water saturated. Four weeks after infection, roots were harvested and snap-frozen in liquid nitrogen until isolation of RNA.

## Staining of fungal structures with Wheat Germ Agglutinin (WGA)-AlexaFluor488

In order to stain fungal and oomycete structures within root tissue, Wheat Germ Agglutinin (WGA) conjugated to AlexaFluor488 (Life Technologies GmbH, Darmstadt, Hesse, Germany) was used enabling staining of cell wall components of fungi and oomycetes, such as *A. euteiches*. Freshly harvested roots were placed in 50% (v/v) ethanol for at least four hours followed by incubation in 20% KOH for 10 min at 90 °C. After washing with distilled water, roots were incubated in 0.1 M HCl for 2 h and then transferred into phosphate-buffered saline (135 mM NaCl, 3 mM KCl, 1.5 mM KH$_2$PO$_4$, 8 mM Na$_2$HPO$_4$, pH 7.4) containing 0.2 μg ml$^{-1}$ WGA-AlexaFluor488 for at least 6 h. Stained roots were analyzed using an epifluorescence microscope AxioImager (Zeiss GmbH, Jena, Thuringia, Germany) using the reflector module 09 (EX BP 450-490/BS FT510/EM LP515). Photographs were taken by an AxioCam (Zeiss) and combined using Adobe Photoshop.

## Monitoring of mycorrhization and *A. euteiches* infection by determination of transcript accumulation using qRT-PCR

Roots frozen in liquid nitrogen were homogenized using mortar and pestle. Total RNA was prepared using the Qiagen RNeasy PlantMiniKit (Qiagen, Hilden, North Rhine-Westphalia, Germany) according to the manufacturer's instruction, and followed by

**Table 1  Primer sequences of genes analyzed with qRT-PCR.**

| Organism | Gene | | Sequence |
|---|---|---|---|
| *M. truncatula* | *Histone3-like* (Medtr4g097170.1) | fwd | 5′-CTT TGC TTG GTG CTG TTT AGA TGG-3′ |
| | | rev | 5′-ATT CCA AAG GCG GCT GCA TA-3′ |
| *M. truncatula* | *MtPT4* | fwd | 5′-ACA AAT TTG ATA GGA TTC TTT TGC ACG T-3′ |
| | | rev | 5′-TCA CAT CTT CTC AGT TCT TGA GTC-3′ |
| *R. irregularis* | *RiBTub* (AF394773) | fwd | 5′-CCA ACT TAT GGC GAT CTC AAC A-3′ |
| | | rev | 5′-AAG ACG TGG AAA AGG CAC CA-3′ |
| *A. euteiches* | 5.8S rRNA (AY683887) | fwd | 5′-TGT CTA GGC TCG CAC ATC GA-3′ |
| | | rev | 5′-AGT GCA ATA TGC GTT CAA CGT TT-3′ |

DNase digestion using Ambion RNase-free DNase (ThermoFisher Scientific, Schwerte, North Rhine-Westphalia, Germany). For real-time qRT-PCR analyses, 1 μg of total RNA was converted into cDNA with Moloney Murine Leukemia Virus Reverse Transcriptase, Point Mutant (Promega, Madison, WI, USA) using oligo(dT)20 primer. The obtained cDNA was diluted 20 times to serve as template for qRT-PCR.

Transcript levels of genes encoding *R. irregularis* β-tubulin (*RiBTub*), *M. truncatula* phosphate transporter4 (*MtPT4*) as well as *A. euteiches*-specific 5.8s rRNA levels were determined according to *Hilou et al. (2014)* using EvaGreen QPCR Mix II (Bio and Sell, Feucht, Bavaria, Germany) and primers listed in Table 1. The Ct-values of the target gene (TG) were normalized to the housekeeping gene *M. truncatula histone3-like* (*MtHIS3L*). The transcript accumulation of a gene from one sample was determined using the mean of three technical replicates. The resulting logarithmic values were converted using the formula $2^{-\Delta Ct}$.

## Statistical analyses

If not otherwise indicated, three independent biological replicates were used for determination of mycorrhization rate and transcript levels. To identify significant differences statistical tests were conducted by Student's *t*-test for pairwise comparisons and one-way-ANOVA with Tukey's HSD test for multiple comparisons. Additionally, two-factorial ANOVA was applied to access interaction effects between the factors genotype and treatment for the data shown in Fig. 1. Standard deviation (SD) is used throughout to indicate variation from the mean.

## RESULTS

### Mycorrhization of core collection of *M. truncatula* accessions

The 32 accessions of the SARDI core collection and the reference line A17 were grown in expanded clay inoculated with *R. irregularis*, and fertilized with Long-Ashton fertilizer containing 3 mM (20%) phosphate instead of 13 mM, which correspond to 100% phosphate. Roots were harvested at three different time points (21, 35 and 50 days post infection [dpi]). Fungal structures in the roots were stained with ink and quantified using the Gridline Intersection method (Figs. 1A–1C). The overall mycorrhization rate increased over time in all accessions, but there were only slight differences in mycorrhization rate

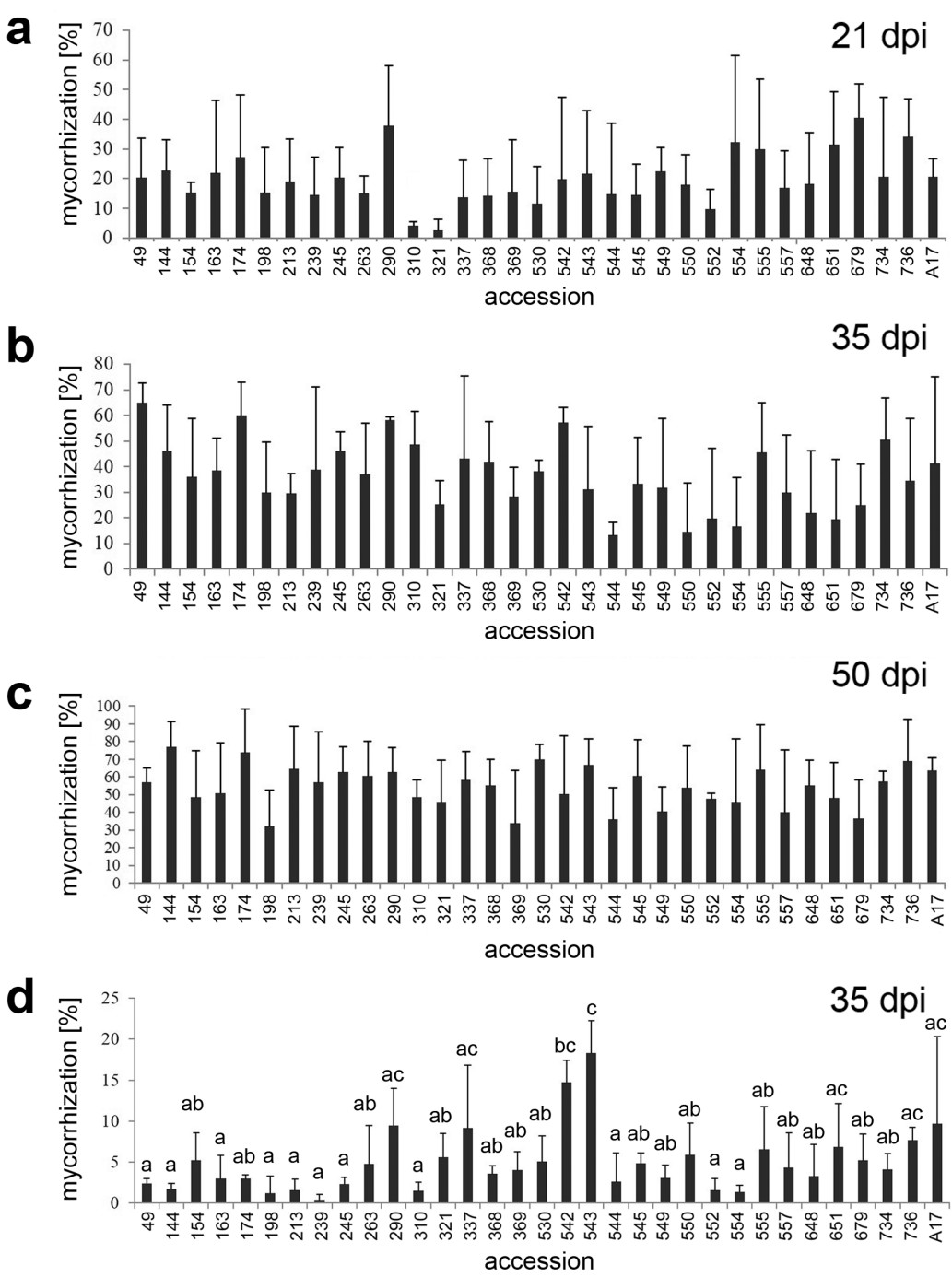

**Figure 1 Mycorrhization rates of the 32 accessions of the core collection and A17.** Plants of the 32 accessions of the core collection and the reference line A17 were inoculated with *R. irregularis* propagated on leek cultures (A, B, C) or were grown in native sandy soil (D). Mycorrhization rate was determined after the time periods indicated. Please note that all line numbers used in the text were abbreviated and contained the last three numerals of their original numbers only (see Table S1). Data are presented as means ±SD ($n = 3$). There were no significant differences in (A, B, C); different letters in (D) indicate significant differences according to one-way-ANOVA with Tukey's HSD test, $p < 0.05$.

between the accessions at all harvest points. The largest differences were found at 21 dpi and ranged from 2.7% (line 321) to 40% (line 679) of root colonization. Comparisons between all lines by ANOVA revealed, however, that there were no significant differences in colonization rates at all three time points of inoculation. The overall development of shoot and root biomass within one accession compared to the other accessions was rather consistent at the different plant ages (not shown). To avoid possible overlay of differences by use of the highly effective inoculum, a native soil was used for an independent experiment (Fig. 1D). This soil had a much lower spore density and higher AM fungi diversity compared to the inoculum of trap cultures. The earliest time point with detectable colonization in this soil was detected at 35 dpi, since the process of mycorrhization was much slower compared to the *R. irregularis* inoculum. The colonization rate ranged from 0.4% to 18.3% (lines 239 and 543, respectively) and showed few significant differences among lines according to one-way ANOVA (Fig. 1D). To assess genotype, treatment and interaction effects, a two-factorial ANOVA was performed using data from all experiments, but no significant interaction effect was detected (Table S2). The high variance in mycorrhization rate of all 32 accessions under all conditions hampered to conclude on clear and reliable differences between the lines.

### Length of primary roots and number of lateral roots in seedlings grown under phosphate-repleted and phosphate-depleted conditions

Since there were only minor differences in the mycorrhization rate and no correlation between mycorrhization rates and plant development, few accessions should be selected for a more detailed analysis. Due to the fact that Pi limitation is a driving force for mycorrhization in *M. truncatula* and at the same time affects the RSA, seedlings of all accessions of the core collection including A17 were grown on plates containing either 100 μM or 3 μM Pi for seven days. There were obvious differences in the RSA monitored in terms of primary root length and lateral root number (Fig. 2, Fig. S1). Under both Pi conditions, no line exhibited significant differences in root length and number of lateral roots compared to A17 according to Students $t$-test with Bonferroni correction. However, comparing root length of seedlings between both Pi conditions, length of primary root decreased significantly in line 163 and increased significantly in lines 321, 368 and 542 under Pi limitation in comparison to high Pi supply (Fig. 2A). There were, however, other lines differing in the number of lateral roots in response to Pi limitation, whereby again the lateral root number changed differently in the lines. Whereas the number in some lines increased, in others it decreased (Fig. 2B). Some lines developed less or no lateral roots under Pi limitation (e.g., 49, 239, 734), whereas others responded to the deficiency with an enhanced number of lateral roots (e.g., 213, 263, A17) or did not exhibit differences (e.g., 555, 736). From these lines showing significantly altered parameters, lines 163, 368 and 542 with changed primary root length under high Pi supply as well as line 213 with highest number of lateral roots under Pi limitation were selected as candidates to analyze its strength of interaction with the AM fungus in more detail. Additionally, line 555 was included, because this line did not change RSA upon Pi limitation and exhibited a stable, but very low number of lateral roots.

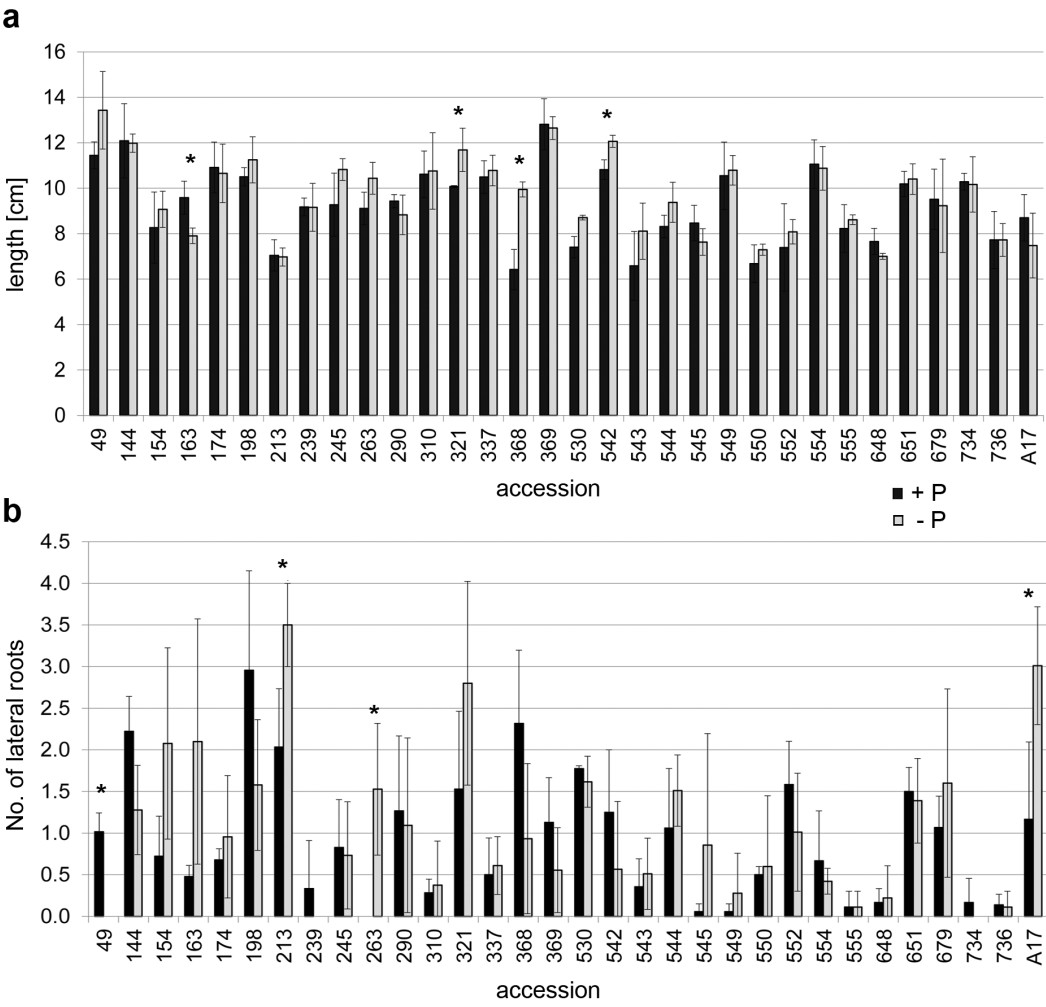

**Figure 2   Root architecture of seedlings.** Seedlings were cultivated on media with 100 μM (black bars) or 3 μM (gray bars) Pi for seven days. (A) Length of primary root, (B) number of lateral roots. Data are presented as means ± SD ($n = 3$ with seven plants each) and were compared between high/low Pi by the Student's $t$ test; * $P \leq 0.05$.

## Mycorrhization of selected accessions under two phosphate regimes

Seedlings of lines 163, 213, 368, 542, 555 and A17 were inoculated with *R. irregularis* and grown either under limited Pi supply (Long-Ashton fertilizer containing 20% phosphate corresponding to 3 mM Pi) or under full Pi supply (Long Ashton fertilizer containing 100% phosphate corresponding to 13 mM Pi) for two weeks. Mycorrhization parameters, such as frequency of mycorrhiza, intensity of mycorrhizal colonization and arbuscule abundance were determined (Figs. 3A and 3B). In comparison to fertilization with 100% Pi, fertilization with 20 % Pi resulted in higher mycorrhizal colonization in all accessions. However, in comparison to the reference line A17, the selected lines did not show differences in all the parameters determined, neither upon Pi limitation nor under full Pi supply. To check the functionality of the symbiosis in more detail, molecular markers, such as the transcript levels of a mycorrhizal fungal gene (*RiTUB*) and of a mycorrhiza-induced plant gene

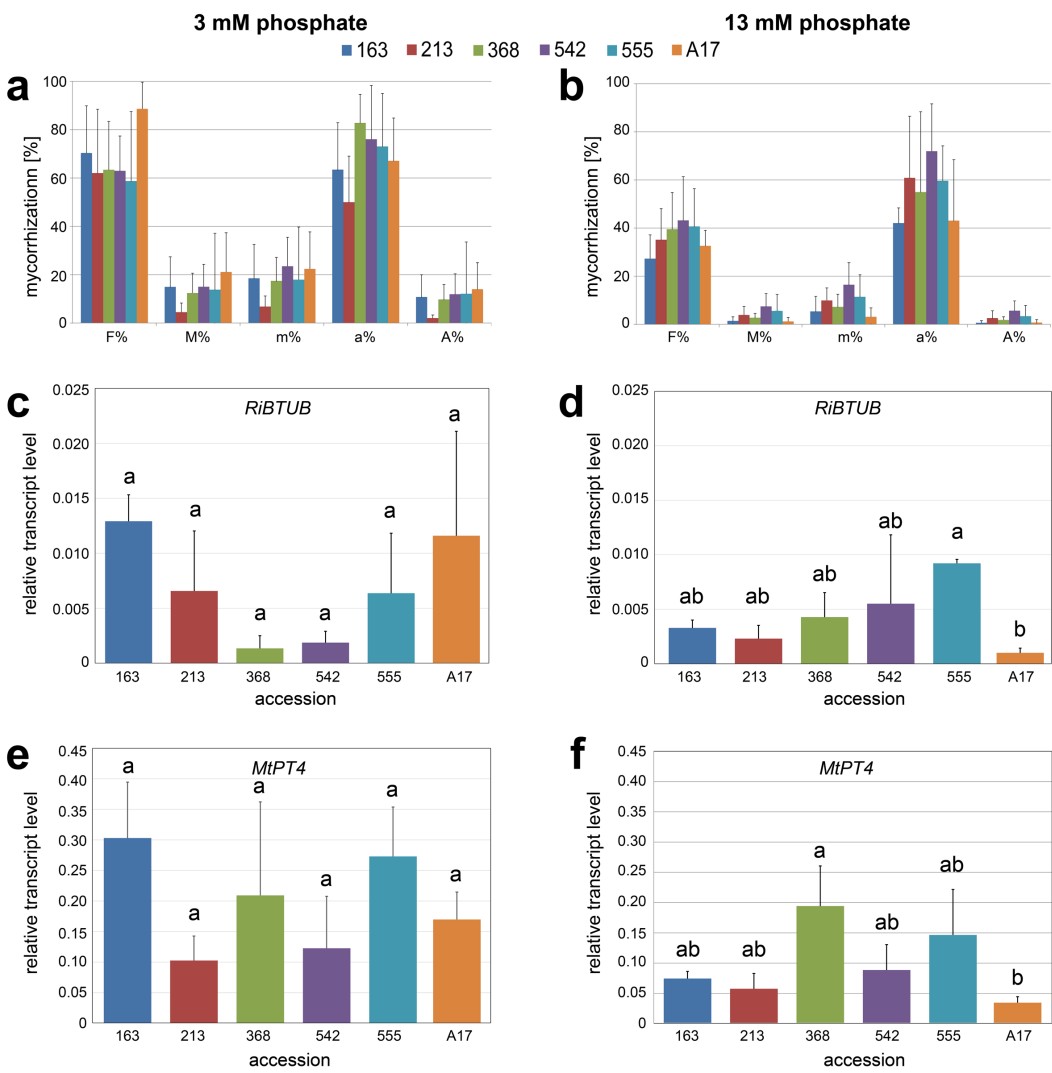

**Figure 3 Mycorrhization of selected accessions.** Accessions No. 163, 213, 368, 542, 555 and reference line A17 were grown in expanded clay, inoculated with *R. irregularis* for two weeks and fertilized either with reduced phosphate levels (20% corresponding to 3 mM Pi, A, C, E) or fully supplied with phosphate (100% corresponding to 13 mM Pi, B, D, F). (A, B) Mycorrhization rate quantified using the method described by *Trouvelot, Kough & Gianinazzi-Pearson (1986)*. F%, frequency of mycorrhiza in the root system; M%, intensity of the mycorrhizal colonization in the root system; m%, intensity of the mycorrhizal colonisation in the root fragments; a%, arbuscule abundance in mycorrhizal parts of root fragments; A%, arbuscule abundance in the root system. Data are presented as means $+$ SD ($n = 5$). There were no significant differences between the ecotypes within each parameter analyzed (tested using one-way-ANOVA with Tukey's HSD test). (C, D) Mycorrhization determined by transcript level of *RiBTUB* in relation to *MtHIS3L*. Data are presented as means $\pm$ SD ($n = 3$). Different letters indicate significant differences (one-way-ANOVA with Tukey's HSD test, $p < 0.05$). (E, F) Functional arbuscules determined by transcript level of *MtPT4* in relation to *MtHIS3L*. Data are presented as means $\pm$ SD ($n = 3$). Different letters indicate significant differences (one-way-ANOVA with Tukey's HSD test, $p < 0.05$).

(*MtPT4*), were determined (Figs. 3C–3F). Upon fertilization with 20% Pi, transcript levels of both marker genes did not show significant differences among the accessions. Upon full Pi supply, however, two lines exhibited significant differences: line 555 showed higher transcript accumulation of *RiTUB* pointing to a higher colonization by *R. irregularis* in comparison to A17. Regarding the *MtPT4* transcripts, line 368 showed significantly higher levels than the reference line. This was accompanied by a high density of arbuscules, which were stained with fluorescently labeled WGA (Fig. S2).

### Infection of selected lines with *A. euteiches*

To check whether the slightly altered mycorrhization phenotype of lines 368 and 555 might be linked to a higher susceptibility towards other soil-born microorganisms, one week old *in vitro* grown seedlings of all selected lines were infected with two strains of *A. euteiches* (Fig. S3). Both strains did not show differences in their infection capability. Roots of all lines appeared to turn brownish, but shoots of susceptible lines did not grow further and developed senescence-like symptoms (lines 163, 213, and 555). When plants were grown in expanded clay and infected with *A. euteiches* (strain GB I1), susceptible lines showed a significant reduction of biomass (Fig. 4A). Line 368 turned out to be highly tolerant towards *A. euteiches* and did not show an impaired fresh weight. Interestingly this was the line, which showed a higher arbuscule abundance determined by *MtPT4* transcript levels. To confirm the infection strength deduced from phenotypic observations, the amount of *A. euteiches* rRNA in infected roots was determined (Fig. 4B). *A. euteiches* rRNA was not detectable in non-infected roots, but in all infected roots. The levels, however, differed drastically—lines previously identified to be susceptible exhibited high levels of *A. euteiches* rRNA, whereas resistant or partially resistant lines showed only low levels of *A. euteiches* rRNA (*Bonhomme et al., 2014*; *Djébali et al., 2009*). This was also visible after staining of hyphae with fluorescently labeled WGA (Fig. S4), where roots of the susceptible lines were heavily stained and that of the partial resistant lines 368 and A17 showed only few labeled hyphae.

## DISCUSSION

Interactions of plants with AM fungi provide several benefits for the plant, on the first place enhanced mineral nutrition (*Smith & Read, 2008*). Therefore, the ability, but also the extent of plants to interact with AM fungi might affect the general plant performance. Genetic diversity present in wild accessions of *M. truncatula* was used to identify putative differences in mycorrhization. The core collection used is part of the South Australian Research and Development Institute (SARDI) collection of *Medicago* spp, which is the largest in the Southern Hemisphere (*Skinner et al., 1999*). Large numbers of molecular markers make it feasible to identify key traits from mapping populations (*Ellwood et al., 2006*; *Ronfort et al., 2006*). In this study, however, no obvious and reliable differences in the mycorrhization rate of these accessions could be found over time. Inoculation with *R. irregularis* and determination of overall mycorrhization rate did not reveal significant differences between the accessions (Fig. 1). This is reminiscent of the data shown by *Schultz, Kochian & Harrison (2010)*, who analyzed eight lines of the SARDI *M. truncatula* collection

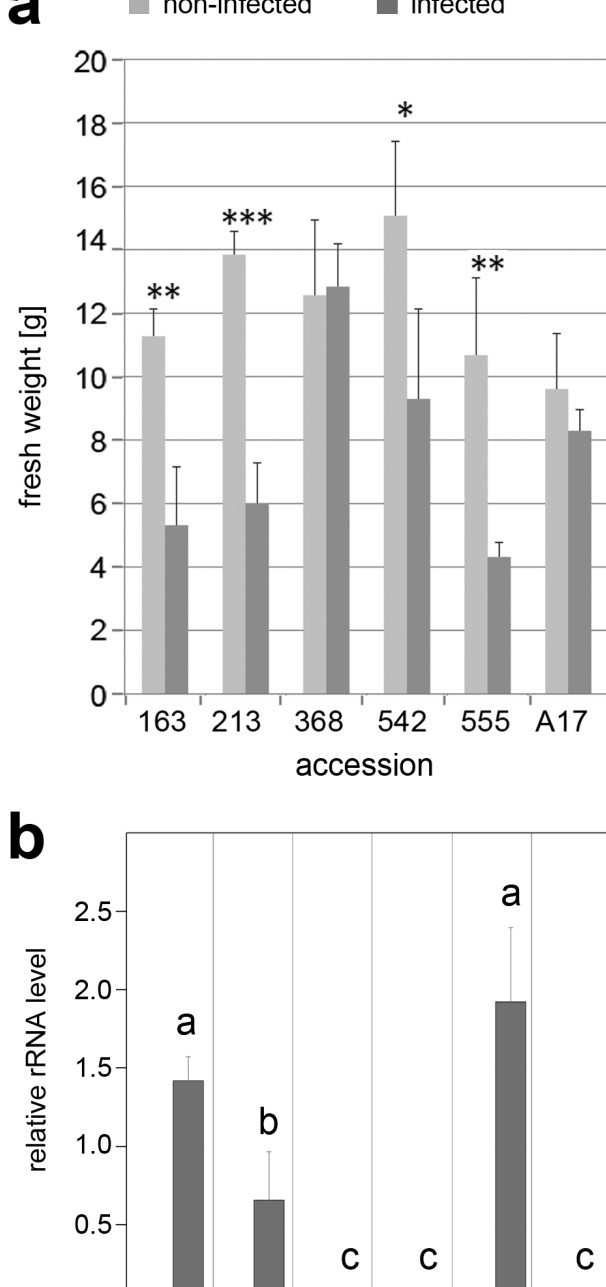

**Figure 4** **Infection of selected accessions with *A. euteiches.*** Seedlings of accessions No. 163, 213, 368, 542, 555 and reference line A17 were grown in expanded clay for two weeks followed by infection with *A. euteiches* strain GB I1 for four weeks. (A) Biomass of plants without (light gray bars) and with (dark gray bars) infection. Data are presented as means ± SD ($n \geq 7$) and were compared between non-infected and infected plants by the Student's *t* test; *$P \leq 0.05$. (B) Infection by *A. euteiches* was determined by level of *A. euteiches* (Ae) rRNA in relation to *MtHIS3L* mRNA. Note the absence of Ae-rRNA in non-infected roots. Data are presented as means ± SD ($n = 3$). Different letters indicate significant differences between the infected plants (one-way-ANOVA with Tukey's HSD test, $p < 0.05$).
in a similar manner by inoculation with *R. irregularis*. They selected different lines of the collection than used in this work, yet two lines (530 referred to as F83005 and 736 referred to as DZA045 in *Schultz, Kochian & Harrison, 2010*) were overlapping to our studies. In both studies (*Schultz, Kochian & Harrison, 2010*), and the present study) it appeared that the process of root colonization by AM fungi is generally variable between individual plants, so that differences between lines are hard to identify. Here, a higher number of replicates would be necessary to improve the probability to find reliable differences between the accessions. Several other crop species, such as pearl millet, barley, maize, durum wheat and onion, showed a large variation between ecotypes regarding the ability for AM fungal symbiosis (*An et al., 2010*; *Baon, Smith & Alston, 1993*; *Galván et al., 2011*; *Kaeppler et al., 2000*; *Krishna et al., 1985*; *Singh et al., 2012*; *Smith, Grace & Smith, 2009*) or in the genetic variation in their capacity to profit from AM symbiosis (*Sawers et al., 2017*), whereas others, such as sorghum, did not show differences in mycorrhization of a large collection of ecotypes (*Leiser et al., 2016*). For the latter example, mycorrhization of various ecotypes showed rather variations in dependence on Pi-availability in the soil. Therefore, mycorrhization of *M. truncatula* accessions under two phosphate fertilization regimes was analyzed using molecular markers. To narrow down the number of accession, five of them were selected according to their RSA response towards full or limited Pi supply.

Under Pi limitation, plants alter their root morphology, topology, and distribution patterns influencing also their interaction with AM fungi (*Bouain, Doumas & Rouached, 2016*). Therefore, RSA was analyzed in terms of length of the primary root and number of lateral roots after growth of seedlings on agar plates either with 100 µM or 3 µM Pi. These two root parameters have been frequently used for determination of Pi deficiency response of various plants, since they are straightforward and easy to assess (*Chevalier et al., 2003*). Under high Pi supply, there were striking differences mainly in the number of lateral roots between all accessions of the core collection (Fig. 2B). There were accessions exhibiting several lateral roots, whereas others did not develop lateral roots within the time frame of the experiment. In response to Pi limitation, only few of the *Medicago* accessions showed significant alterations in length of primary roots, among them three lines exhibited an increase and one line a decrease in primary root length. In comparison to high Pi supply, the number of lateral roots was also changed only in few accessions, whereby three lines showed a significant increase in lateral root number. One line showed a significant decrease in the number of lateral roots, because it did not develop any lateral roots under Pi limitation. Pi deficiency has been shown to reduce the growth of primary roots and to enhance lateral root formation and the length and density of root hairs in many plant species (*Desnos, 2008*; *López-Bucio, Cruz-Ramírez & Herrera-Estrella, 2003*), among them *Arabidopsis thaliana* Col-0 (*Péret et al., 2014*). However, not all ecotypes of *A. thaliana* show these typical features. Out of 73 accessions nearly 25 % did not respond to Pi limitation by shorter primary roots (*Chevalier et al., 2003*). Moreover, Pi limitation has opposite effects in monocots showing promotion of primary root growth and inhibition of lateral root formation (*Li et al., 2012*; *Sun et al., 2014*). Other plant species, such as white lupine (*Lupinus albus*), are able to develop so-called cluster roots, which are covered by large numbers of dense root hairs (*Lambers et al., 2006*) and present another evolutionary

adaptation mechanism in response to Pi starvation. This points to a genetically determined root architectural response to low Pi conditions for better acquisition of Pi through root morphology and physiological adjustment, which cannot be generalized for all species (*Péret et al., 2014*).

Mycorrhization levels were determined in the five selected *Medicago* accessions using two different Pi fertilization regimes. Again, the differences in mycorrhization rate between the accessions were not significantly different, mainly due to the high variation between single plants of one ecotype. However, the dependence of mycorrhization rate on Pi supply was obvious. Under full Pi supply (13 mM Pi) frequency and intensity of mycorrhizal colonization were reduced in comparison to both parameters determined from plants grown under Pi limitation (3 mM Pi). It is well known that symbiosis establishment is promoted under Pi deficiency conditions and is limited under full Pi supply (*Andreo-Jimenez et al., 2015*). Even addition of Pi to mycorrhizal plants leads to a reduction of colonization preceded by rapid repression of symbiotic gene expression as shown for petunia (*Breuillin et al., 2010*). Under full Pi supply (limited mycorrhization), two accessions (lines 555 and 368) exhibited, however, slightly enhanced mycorrhization. Roots of line 555 harbored significantly more fungal material (determined by transcript accumulation of fungal housekeeping gene encoding β-tubulin) and line 368 harbored more active arbuscules as indicated by the significantly enhanced transcript levels of *MtPT4*. Transcript levels of *MtPT4* are indicative for the arbuscule abundance including their functionality, because it is directly linked to one key feature of the mycorrhizal symbiosis, namely the transport of phosphate (*Isayenkov, Fester & Hause, 2004*; *Javot et al., 2007*). Since the differences were detectable only upon full Pi supply, the question raised, whether these lines might be more susceptible also to other microorganism infecting roots. Both accessions behaved, however, differently in respect to infection with *A. euteiches*: Whereas line 555 was highly susceptible, line 368 was partially resistant. This was visible not only in a plate assay performed with two strains of *A. euteiches*, but also in adult plants grown and infected in pots. Next to the decrease in biomass of all susceptible lines, the amount of *A. euteiches* rRNA was increased in these roots showing unequivocally the different infection levels of the selected lines (Fig. 4). Differences in the colonization of roots by *A. euteiches* were additionally visible after WGA staining—the highly susceptible lines exhibited a rather dense hyphal network. Among the tested lines, line 368, 542 and A17 belong to the (partial) resistant accessions as demonstrated by *in vitro* inoculation assays previously published (*Bonhomme et al., 2014*; *Djébali et al., 2009*). This points to the fact that mycorrhization phenotype and pathogen susceptibility of *M. truncatula* are not correlated, neither positively nor negatively.

## CONCLUSION

Interaction of plants with AM fungi is beneficial for plants not only for better supply with nutrients but also as bioprotective agent against root and foliar diseases (*Hilou et al., 2014*; *Jung et al., 2012*). Genetically diverse *Medicago* accessions with clearly different responses towards AM fungi would help to identify underlying genes regulating this symbiotic

association. Genetic diversity among the accessions of the SARDI collection in regard to resistance against *A. euteiches*, in drought-related traits and in morphological traits, such as plant height, trichome density and flowering time, have been successfully used to identify candidate genes by a Genome-Wide Association Study (GWAS) (*Bonhomme et al., 2014*; *Kang et al., 2015*; *Stanton-Geddes et al., 2013*). Thereby, the whole genome sequence and single nucleotide polymorphism (SNP) information for 288 inbred accessions provided by the HapMan project (http://www.medicagohapmap.org/) represent tremendous resources for GWAS. Although there are differences in mycorrhization among the tested accessions, the high standard variations will, however, hamper such an identification of candidate genes either by QTL analyses or GWAS. Here, a higher number of biological replicates or much more strongly contrasting phenotypes might be useful to identify genes that control basal processes in root development in response to environmental impacts. It is tempting to speculate that the differences in RSA and/or in plant's response to Pi limitation might be traits to be used for identification of regulatory genes.

# ACKNOWLEDGEMENTS

Dr. Anne Moussart and Prof. Phillip Franken are kindly acknowledged for providing *Aphanomyces euteiches* (Drechs.) strain AERB84 and strain GB I1, respectively. We thank Jane Gohlisch and Carolin Delker for helping in determination of mycorrhizal parameters and performing statistical analyses, respectively.

## Funding

This work was supported by a grant from the Leibniz-Association (SAW-PAKT "Chemical Communication in the Rhizosphere") and by the BRAVE project funded by the ERASMUS MUNDUS Action 2 program of the European Union. The funders had no role in study design, data collection and analysis, decision to publish, or preparation of the manuscript.

## Grant Disclosures

The following grant information was disclosed by the authors:
Leibniz-Association.
ERASMUS MUNDUS Action 2 program of the European Union.

## Competing Interests

Bettina Hause is an Academic Editor for PeerJ. The other authors declare there are no competing interests.

## Author Contributions

- Dorothée Dreher conceived and designed the experiments, performed the experiments, analyzed the data, wrote the paper, prepared figures and/or tables, reviewed drafts of the paper.
- Heena Yadav conceived and designed the experiments, performed the experiments, analyzed the data, prepared figures and/or tables, reviewed drafts of the paper.

- Sindy Zander performed the experiments, reviewed drafts of the paper.
- Bettina Hause conceived and designed the experiments, analyzed the data, wrote the paper, prepared figures and/or tables, reviewed drafts of the paper.

### Data Availability

The raw data have been uploaded as Supplemental Files.

### Supplemental Information

Supplemental information for this article can be found online at http://dx.doi.org/10.7717/peerj.3713#supplemental-information.

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
