# Peer review of "Is there genetic variation in mycorrhization of Medicago truncatula?"

_PeerJ, doi:10.7717/peerj.3713_

## Round 0.1 · original submission · Major Revisions

Both reviewers considered the manuscript valuable to the community and a fully agree. They made a number of important points that need to be addressed to increase the impact of the manuscript.

The major revision that is required is to extend the statistical analysis of the data. I completely agree with reviewer 1 that comparison to a reference accession is not appropriate way of performing the analysis, the comparison has to be among the genotypes. In addition, the Broad Sense Heritability should be calculated.

I also agree that the number of replicates is rather low for this type of study. However, as increasing the number would be an effort not proportional to the gain in knowledge, therefore only the discussion of this issue is required.

In addition, both reviewers made a number of relatively minor points, mainly concerning presentation, that need to be addressed.
I believe these revisions are relatively straightforward and I am looking forward to receiving the revised version of your manuscript.

Reviewer 1 ·

Basic reporting

The manuscript by Dreher et al. entitled “Is there genetic variability in mycorrhization of Medicago truncatula?” addresses an important and up-to-date topic that is a prerequisite for association genetics studies. The Authors evaluated the mycorrhizal levels of 33 Medicago truncatula accessions at 3 time points with traditional staining techniques and then focused on a subset of those accessions evaluating also gene expression, root system architecture (RSA) responses to phosphate levels and interaction with a pathogen. Since they do not find a significant difference with the reference genome (A17) through a t-test approach, their main conclusion is that there is not enough phenotypic variability among accessions to proceed with further association genetics studies (for example GWAS).
Notwithstanding the fact that the Authors are mainly drawing negative conclusions, I think some more statistical tests should be done (Anova) and the Authors should discuss the influence of standard deviation and sample variability in their setup. This would allow to better read the data they produced and could constitute a valuable resource for the community.

Experimental design

In general the experiment design was clearly described but some additional statistical tests could help in describing the data more accurately and get to the right conclusions.

In particular:

Line 24 – Statistical tests should be run on the whole panel and not just compared to A17. For example, Anova test and post-hoc test (Bonferroni) would allow to say if there are or not differences among genotypes. It could also be used to see if there are interaction effects (genotype ~ day). Also Broad Sense Heritability could be calculate to have an idea of which portion of the variance can be explained by genotype.

Line 231: In my honest opinion, to investigate genetic diversity within one species the focus should not be the comparison with the reference genome (A17) but the existence or not of variability among the all accessions and how much of this variance is due to the genotype.

Validity of the findings

Minor comments:
Line 40 –Staining could not be enough to detect real mycorrhizal differences among accessions. For example MtPT4 levels could tell if the functionality of the nutrient exchanges are different or not. Authors should discuss it.

Revise line 49-51: sentence is not complete.

Line 71: cite also latest version of the genome (Tang et al., 2014)

Line 110-111: Please cite other works linking AM marker genes and RSA in Lotus (Volpe et al., 2016).

Line 225: it is not clear what the 20% refers to. The whole Long Ashton is diluted 1:5?

Line 244: I don’t think that 100 uM of phosphate could be considered a high concentration. Better use phosphate-depleted and repleted.

Fig. 3a: Better show in the figure the final phosphate concentration and not the percentage so that the reader can get the information in an easier way. Using the same colours to indicate the same genotypes across panels will allow to compare between different panels in a straight way.

Line 325: Every biological process is subjected to variability. In my honest opinion, Authors should mention in the discussion that a higher number of replicates could allow to better point at differences between accessions.

Line 327: add also citation for the last work from Paskowski group on maize (Sawers et al., 2017, New Phytologist).

Discussion from line 362 on it is not flowing and it is hard to read. Moreover it seems that in this section the Authors are describing how some accessions respond differently to mycorrhization under high phosphate (line 372) or show higher MtPT4 transcript (line 375): in both cases, I see some contradictions with results described in the abstract (line 41). Is it enough to say that the trait cannot be used as a trait to access genetic markers (line 42)?

Line 396: in my honest opinion, you don’t need “unique responses” to map those trait to the genome, whereas a certain degree of diversity easy to quantify and reproducible.
Line 405: The conclusion is too strong: the differences among accessions that you show in Fig.1a and Fig.1b is not so minor. What it is probably not good for genetic association studies is the standard deviation but it is also true that 3 biological replicates per each accession (each consisting of a single individual) are probably a too small sample. Authors should discuss this point.

Additional comments

As a general comment, I found the results valuable being shared with the community but I really think that some more statistics should be performed to support the negative conclusions. Moreover in my opinion three biological replicates consisting of one plant each will always show big variation in any possible trait and no strong (especially negative) conclusions should be taken out of it.
Finally Broad Sense Heritability would constitute a better proxy to investigate plant panel diversity and possibility to make further genetic studies.
For sure, the manuscript is presenting all the required information and raw data so that every step can be redone and analyzed by the reader.

·

Basic reporting

no comment

Experimental design

no comment

Validity of the findings

no comment

Additional comments

The manuscript ‘Is there genetic variation in mycorrhization of Medicago truncatula?’ by D. Dreher et al. is well written and clearly answered the question that there's only minor difference in mycorrhization of Medicago accessions.
Some minor points:
1. In page 18 line 231, the conclusion is ‘Compared to the reference genotype A17, there were no significant differences in colonization rate’. Is the difference of mycorrhization rate between line 310 (or line 321) and A17 at 21 dpi not significant using Student’s t test? Give details for how to perform ‘Student’s t-test including a Bonferroni-correction for multiple tests’ in line 217.
2. Include ref. in line 302 for the Aphanomyces euteiches tolerant and susceptible lines in Medicago.
3. In Figure 1 legend ‘Mycorrhization rates of the 32 accessions of the core collection and A1.’, A1 should be A17.

---

## Round 0.2 · accepted · Accept

Thank you for taking the reviewers´ comments on board, they were all carefully addressed and the manuscript is thus suitable for publication.